# Wheel Defect Detection Using a Hybrid Deep Learning Approach

**DOI:** 10.3390/s23146248

**Published:** 2023-07-08

**Authors:** Khurram Shaikh, Imtiaz Hussain, Bhawani Shankar Chowdhry

**Affiliations:** 1Department of Electronic Engineering, Mehran University of Engineering and Technology, Jamshoro 76062, Pakistan; khuram.shaikh@admin.muet.edu.pk; 2Department of Electrical Engineering, National University of Sciences and Technology, Islamabad 44000, Pakistan; 3NCRA-Condition Monitoring Lab, Mehran University of Engineering and Technology, Jamshoro 76062, Pakistan; bhawani.chowdhry@faculty.muet.edu.pk

**Keywords:** wheel defects, deep learning, wheel flats, false flange, nonlinear dynamics, MLP

## Abstract

Defective wheels pose a significant challenge in railway transportation, impacting operational performance and safety. Excessive traction and braking forces give rise to deviations from the intended conical tread shape, resulting in amplified vibrations and noise. Moreover, these deviations contribute to the accelerated damage of track components. Detecting wheel defects at an early stage is crucial to ensure safe and comfortable operation, as well as to minimize maintenance costs. However, the presence of various vibrations, such as those induced by the track, traction motors, and other rolling stock subsystems, poses a significant challenge for onboard detection techniques. These vibrations create difficulties in accurately identifying wheel defects in real-time during operational activities, often resulting in false alarms. This research paper aims to address this issue by using a hybrid deep learning-based approach for the accurate detection of various types of wheel defects using accelerometer data. The proposed approach aims to enhance wheel defect detection accuracy while considering onboard techniques’ cost-effectiveness and efficiency. A realistic simulation model of the railway wheelset is developed to generate a comprehensive dataset. To generate vibration data in various scenarios, the model is simulated for 20 s under different conditions, including one non-faulty scenario and six faulty scenarios. The simulations are conducted at different speeds and track conditions to capture a wide range of operating conditions. Within each simulation iteration, a total of 200,000 data points are generated, providing a comprehensive dataset for analysis and evaluation. The generated data are then utilized to train and evaluate a hybrid deep learning model, employing a multi-layer perceptron (MLP) as a feature extractor and multiple machine learning models (support vector machine, random forest, decision tree, and k-nearest neighbors) for performance comparison. The results demonstrate that the MLP-RF (multi-layer perceptron with random forest) model achieved an accuracy of 99%, while the MLP-DT (multi-layer perceptron with decision tree) model achieved an accuracy of 98%. These high accuracy values indicate the effectiveness of the models in accurately classifying and predicting the outcomes. The contributions of this research work include the development of a realistic simulation model, the evaluation of sensor layout effectiveness, and the application of deep learning techniques for improved wheel flat detections.

## 1. Introduction

The railway wheelset plays a vital role in railway transportation, ensuring the safe and comfortable operation of the vehicles. The conical shape of the wheel tread is designed to maintain optimal performance. However, excessive traction and braking forces at the wheel–rail interface can cause the wheel’s exterior perimeter to deviate from its desired conical shape. This either results in a flat or more conical surface of the wheel tread. This deviation leads to increased vibrations, noise, and a decrease in ride comfort and operational safety. Moreover, wheel defects contribute to the accelerated growth of cracks on the rail tracks, ultimately leading to premature failure of the rail system [1]. Statistical analysis of mechanical components of trains between 2004 and 2007 revealed that wheelset faults accounted for 44.7% of train accidents, making them the most significant cause [2]. Therefore, early detection of defective wheels is crucial. To achieve this, an automated approach capable of accurately distinguishing between healthy and damaged wheels needs to be developed. Consequently, there is a great interest among railway administrations and rolling stock operators in finding effective methods for early detection and identification of wheel flats [3].

Rolling stock inspection is normally performed at fixed intervals and carried out periodically in workshops. Periodic inspection is expensive due to the unavailability of rolling stock during maintenance time. Moreover, it is also inefficient, which usually results in over- or under-maintenance of the components [4]. In the last few decades, various techniques have been employed in the railway industry, and various monitoring approaches have been proposed to automatically inspect wheel conditions. Most of these techniques are based on the concept that the wheel–rail interaction forces increase in a defective wheel [5,6]. The most popular techniques which have gained plenty of attention from railway researchers are the onboard techniques and wayside measurement techniques [7]. 

The wayside inspection involves the installation of various sensors along the railway track at specific locations, allowing for the evaluation of all passing wheels [8,9]. These techniques utilize vibration, acoustic, image detection, and ultrasonic technologies to monitor the condition of the trains [10,11,12]. Hot axle box detectors (HABDs) are commonly employed in wayside monitoring systems to identify faulty overheating axle bearings by using thermal imaging. However, HABDs are unable to detect damage at early stages, as minor faults do not typically cause a noticeable increase in temperature. Additionally, HABDs are prone to false alarms triggered by environmental conditions [13]. Trackside acoustic array detectors (TAADS) utilize arrays of microphones to capture the audible noise produced by passing axle bearings. However, the operating frequency range of these microphones (normally 22–44 kHz) makes them susceptible to errors resulting from background noise [14]. To enhance the accuracy of detection, many modern wayside monitoring systems incorporate machine learning and deep learning algorithms [5,15]. For example, in [5], a multi-sensor data fusion approach combined with an unsupervised early damage detection methodology has been proposed, capable of automatically distinguishing between defective and healthy wheels, particularly considering small flat sizes. This methodology relies on the evaluation of acceleration and shear time histories recorded on the rails during the passage of traffic loads. In [15], an improved YOLOv3 framework is developed for rail wheel surface defect detection, achieving classification detection of four types of wheel tread defects with an average mean average precision (MAP) accuracy of 0.92. Various types of sensors, including strain gauges and accelerometers, are utilized to capture input signals for detecting wheel flats.

However, wayside monitoring systems are often costly due to the requirement for multiple sensors and high-end computing capabilities for comprehensive diagnosis and effective wheel condition monitoring. For instance, in the Swedish railway network, the wayside equipment for monitoring rolling stock consists of almost 200 wayside inspection devices [16]. Moreover, this method also requires detailed information about a target vehicle (e.g., number of axles and type of wagon) for accurate condition monitoring [17]. The high cost and maintenance issues associated with this method limit its widespread use. Additionally, the deployment and security of wayside equipment are important considerations that need to be addressed.

Due to the challenges associated with wayside monitoring techniques, several onboard techniques have been proposed in the literature [18,19,20,21,22,23]. One such technique, presented in [10], detects wheel flat defects by measuring the vertical acceleration on the axle box. This algorithm operates in the time domain and can identify the presence of wheel flats at an early stage and estimate their severity. The results demonstrate that the proposed wheel flat index is effective for detecting small flats and estimating their severity. In [18], multiple models and a fuzzy logic-based technique are introduced for detecting conicity in railway wheelsets. This method indirectly identifies the conicity condition by analyzing the lateral acceleration of the wheelset. Another onboard detection method proposed in [7], known as the angle domain synchronous averaging technique (ADSAT), utilizes vertical axle-box vibration acceleration (ABVA) to monitor the conditions of axle-box bearings. The results indicate that this method not only achieves better detection than traditional methods but also mitigates the influence of background noise. In [19], a lightweight 1D convolutional neural network (CNN) architecture, guided by Bayesian optimization, is presented for wheel flat (WF) detection using car body accelerations. Additionally, model-based onboard techniques are proposed in [20,21,22,23] to detect wheelset conditions under different environmental conditions, utilizing axle vibration and gyroscopic data for onboard diagnosis.

The advantage of on-board monitoring systems is that the wheel is monitored continuously and not only when the vehicle passes a trackside monitoring site. This allows for the timely detection of emerging wheel defects [10], thus, allowing immediate action to perform maintenance after the formation of a wheel flat, without requiring a visit to a trackside monitoring site [10,24]. Furthermore, if the onboard monitoring system provides positioning, the occurrence of a wheel defect can be linked to a position on the track. The track at this position can then be inspected and, in cases where a track defect is identified, appropriate maintenance actions can be issued to avoid further damage to the rail and other passing vehicles [24]. The early detection of wheel flats can also be correlated with braking occurrences, if the braking system is monitored, to develop strategies for reducing their frequency [25]. However, onboard detection methods require equipping all wheels with sensors for comprehensive diagnosis and effective wheel condition monitoring, which can be costly and pose maintenance challenges.

Onboard detection techniques face a substantial challenge due to the presence of diverse vibrations originating from multiple sources within the railway system. These vibrations, generated by the track, traction motors, and various rolling stock subsystems, introduce significant complexity and pose obstacles to the accurate identification of wheel defects in real-time during operational activities [7,26]. The intricate nature of these vibrations makes it difficult to distinguish between genuine wheel defects and false alarms, leading to potential disruptions and unnecessary maintenance interventions. Addressing the challenge of vibration interference requires innovative approaches that can effectively separate the signals related to wheel defects from the surrounding vibrations. By employing machine learning algorithms, it becomes possible to enhance the accuracy of onboard detection systems and mitigate the impact of background vibrations. These techniques aim to extract relevant features and patterns associated with wheel defects, enabling reliable and real-time identification even in the presence of complex vibration environments [27].

This paper presents a novel technique to address the challenges faced by onboard techniques. The proposed technique utilizes axle vibration data exclusively, thereby reducing the reliance on multiple sensors. Moreover, the proposed technique incorporates a hybrid deep learning approach, which plays a crucial role in mitigating the influence of background vibrations. The utilization of hybrid deep learning algorithms allows for enhanced detection performance by effectively distinguishing between genuine wheel defects and the vibrations induced by the track, traction motors, and other rolling stock subsystems. By harnessing the power of deep learning, the proposed technique aims to overcome the limitations and false alarms commonly encountered in onboard detection systems.

A significant novelty of this research is the automatic detection of defective wheels at a very early stage, representing a substantial improvement in the effectiveness of the proposed method and facilitating its full implementation in real-world applications. The research work’s key contributions are as follows:(a)Development of a realistic simulation model of the railway wheelset to generate a comprehensive dataset.(b)Evaluation of the effectiveness of the proposed method considering a minimalist layout of sensors.(c)Enhancement of wheel flat detection through the application of hybrid deep learning technique.

## 2. Methodology

The methodology section of this research paper outlines the approach used to develop a deep learning model for the detection of defects on railway wheelsets using vibration measurements. This section describes the data collection process, the deep learning architecture employed, the training procedure, and the model evaluation methodology. By leveraging the inherent capabilities of deep learning, the proposed methodology aims to automatically learn relevant features directly from the raw vibration measurements, eliminating the need for explicit feature extraction. The following subsections provide a detailed overview of the methodology, highlighting the steps taken to train and evaluate the deep learning model accurately.

### 2.1. Development of a Realistic Simulation Model

A nonlinear wheelset simulation model presented in [21,22,23,28] is used to develop the simulation model in Simulink to mimic the behavior of the actual wheelset dynamics. The model is modified to consider all the disturbances that are faced by an actual railway vehicle (e.g., irregularities in track in the lateral direction, variation in gauge and track geometry) to generate data that are close to real-time scenarios. Only the lateral and yaw dynamics of the system are considered, which are most affected by the wheel defects. The model is described in Equations (1)–(5).
(1)$$\ddot{y}=\frac{2}{m_{w}}\left(\Psi \frac{F_{a}}{\gamma }-\frac{\dot{y}}{V_{X}}\frac{F_{a}}{\gamma }\right)$$
(2)$$\ddot{\Psi }=\frac{1}{I_{w}}\left(\frac{2y_{g}y_{t}L_{g}}{r_{0}}\frac{F_{a}}{\gamma }-\frac{2{\mathrm{yL}}_{g}\lambda_{w}}{r_{0}}\frac{F_{a}}{\gamma }-\frac{2{\dot{\Psi }L}_{g}{}^{2}}{V_{X}}\frac{F_{a}}{\gamma }-K_{w}\Psi \right)$$
(3)$$\gamma =\sqrt{{\left(\frac{L_{g}\dot{\Psi }}{V_{X}}+\frac{\lambda_{w}\left(y-y_{t}-y_{g}\right)}{r_{0}}\right)}^{2}+{\left(\frac{\dot{y}}{V_{X}}-\Psi \right)}^{2}}\;$$
(4)$$\mu =u_{0}[\left(1-A\right)e^{\left(-B\gamma V_{X}\right)}+A]\;$$
(5)$$F_{a}=\frac{2N\mu }{\pi }\left[\frac{k_{A}\varepsilon }{1+{\left(k_{A}\varepsilon \right)}^{2}}+\arctan\left(k_{S}\varepsilon \right)\right]$$

The simulations are in different conditions to develop a comprehensive dataset. The following scenarios are considered in the development of the dataset. 

Variation in speed $(V_{X}$): In previous similar studies [18,19,20,21,22,23,24] speed was kept constant to detect the anomalies. However, in real-time scenarios, speed is not constant. Therefore, in this study, data are generated at variable speeds (25 km/h, 50 km/h, 75 km/h, 100 km/h, 120 km/h, and 150 km/h). 

Variation in track conditions: Track condition also changes with time and location. Therefore, the data are generated by varying the track parameters ($\mu ,\;k_{A},\;\varepsilon ,\;\mathrm{and}\;k_{S}$). 

Variation in wheelset condition: Wheel profile plays an important role in the safe and reliable operation of railway vehicles. Due to wear and tear during the operation wheel profile is changed. The variation in the wheel profile is incorporated during the simulation by varying wheel conditions ($\lambda_{w}$). 

Track disturbances: The creep forces generated at the wheel–rail interface are directly affected by the track irregularities, such as lateral variation and variation in gauge. Therefore, these disturbances are also considered in this study.

The data collection process is depicted in Figure 1. A simulation model of a nonlinear railway wheelset is developed in MATLAB/Simulink. Varying forward speed, varying wheel conditions, and varying track conditions are given as input to the model. Practically, a 3-axis accelerometer is placed on the left side of the axle (opposite side of the traction motor) to measure axle vibration. A preprocessing unit is used to remove any bias in the vibration data and to extract vibrations in lateral direction. Simulations are run for 20 s in each of the scenarios, including one non-faulty scenario and six faulty scenarios, to develop the dataset. Some of the results are given in Figure 2. It is quite evident from Figure 2 that the frequency and the amplitude of the lateral acceleration are directly affected by the changes in the speed as well as by the wheel profile variation. An accelerometer is placed on the axle of the wheelset to capture lateral vibrations during the simulation process. A total of 200,000 data points are generated in each scenario which is archived in CSV format for later use.

### 2.2. Hybrid Detection Architecture 

The overall block diagram for defect detection is shown in Figure 3. The defect detection architecture proposed in this research paper combines an MLP as a feature extractor with a machine learning model for the final classification of conicity values. The input features used in this architecture are derived from vibration data only. This is one of the main contributions of this work because in the presence of disturbances and variable parameters it is difficult to detect wheel defects using vibration data only. 

To ensure effective feature extraction, the MLP undergoes a pre-training phase, as shown in Figure 4a. During pre-training, the model is initialized with appropriate weights and biases, enabling it to extract meaningful patterns from the input data. This initialization process sets a foundation for the MLP to learn and capture relevant features that are crucial for accurate defect detection in railway wheelsets.

### 2.3. MLP—Fully Connected Network

The model presented in this research paper is designed for defect detection in railway wheelsets using vibration data. The architecture, shown in Figure 5, consists of an MLP as a feature extractor and a subsequent machine-learning model for the final classification of conicity values. The feature extraction process begins with the MLP, which includes three parallel dense networks. Each network has multiple dense layers with decreasing numbers of units, incorporating the GELU activation function. Regularization is applied to the first dense layer of the first parallel network using a kernel regularizer.

During training, the model is initially pre-trained to initialize the weights and biases. This pre-training step helps the MLP extract meaningful patterns from the vibration data, facilitating effective feature extraction. Once the MLP is pre-trained, the last classification layer is removed. Instead, the layer where the outputs of all three branches are concatenated is used to extract the features learned by the MLP. This concatenated layer combines the features extracted from each parallel network with the input layer, providing a comprehensive representation of the input data. A training summary of the MLP is given in Table 1.

These extracted features serve as the inputs to the subsequent machine learning model, which performs the final classification of conicity values. The machine learning model utilizes the extracted features to make accurate predictions regarding the severity of defects in railway wheelsets. By removing the last classification layer and extracting features from the concatenated layer, the model focuses on capturing and utilizing the learned representations of the input data. This approach enhances the effectiveness of defect detection by leveraging the discriminative power of the extracted features for classification purposes.

### 2.4. Hyperparameters 

The details of the hyperparameter are given in Table 2. The model uses a Gaussian error linear unit (GELU) as an activation function with a categorical cross-entropy loss function and Adam optimizer. 

#### 2.4.1. GELU Activation Function

The GELU activation function outperforms ReLU in the presented architecture due to its smoothness, improved representation learning capabilities, and ability to alleviate the vanishing gradient problem. These advantages enable more stable training, enhanced feature extraction, and improved accuracy in defect detection for railway wheelsets. The GELU activation function is defined as follows in Equation (6): (6)$$GELU\left(x\right)=0.5x\cdot \left(1+\tanh\left(\sqrt{\frac{2}{\pi }}\cdot \left(x+0.044715x^{3}\right)\right)\right)$$

#### 2.4.2. Categorical Cross Entropy Loss Function

Categorical cross entropy is utilized as the loss function for the final classification model. This choice is motivated by the nature of the defect detection task, where the goal is to classify conicity values into different categories or severity levels. Categorical cross entropy is well-suited for multi-class classification problems, providing a measure of the dissimilarity between the predicted conicity values and the true labels. By minimizing this loss function, the model is encouraged to accurately classify the severity of defects in railway wheelsets.
(7)$$Loss=-\sum_{i=1}^{\begin{array}{c} output\\ size \end{array}}y_{i}\cdot log\hat{y_{i}}$$

#### 2.4.3. Adaptive Moment Estimation (ADAM) Optimizer

The ADAM optimizer is a widely used optimization algorithm for training neural networks. It offers several advantages that contribute to its popularity. ADAM utilizes adaptive learning rates, adjusting the learning rate for each parameter individually based on the gradients’ history. This adaptive nature allows for faster convergence and efficient parameter updates. The inclusion of momentum helps accelerate optimization by maintaining a running average of past gradients. ADAM also handles sparse gradients effectively, which is common in deep learning models. Additionally, it incorporates L2 regularization, preventing overfitting and improving generalization. Overall, these features make ADAM a versatile and effective optimizer for neural network training.

## 3. Results

### 3.1. Training Procedure

The entire training procedure for the proposed algorithm consists of pretraining an MLP for feature extraction and then feeding these feature vectors to ML model training. The steps are shown in Table 3.

### 3.2. Model Performance

The results are gathered by following a training procedure where the MLP network is first pre-trained. Then, using the trained model, the last classification layer is removed to extract features. These features are then fed to ML models one by one, and a comparison is drawn. Each model is analyzed for its accuracy, precision, recall, and F1 score. The performance metrics are summarized in Table 4.

Figure 6 shows the performance comparison of MLP, SVM, RF, DT, and k-N. The MLP-RF shows the highest detection accuracy, precision, and F1 Score.

### 3.3. Results Interpretation

The performance results in the previous section show four types of performance metrics, namely accuracy, precision, recall, and F1 score. The results are summarized in Table 5. 

### 3.4. Classification Results for Conicity Values

Figure 7 shows the error matrix (or confusion matrix) for predicted and ground truth results. Starting from MLP, the accuracy of valid conicity value (c = 0.15) is 82%, while for MLP-RF it is 99%, for MLP-DT it is 98.9%, for MLP-kNN it is 94.2%, and for MLP-SVM it is 79.3%. The confusion matrix is plotted for the unseen test data.

## 4. Conclusions

In conclusion, this research paper presents the implementation of a hybrid deep learning-based defect detection system. The system utilizes a multilayer perceptron (MLP) for feature extraction, which is pre-trained on collected data. Multiple machine learning models, including support vector machine (SVM), random forest (RF), decision tree (DT), and k-nearest neighbors (k-NN), are employed for classifying conicity target values. The performance of different models in detecting wheel defects was evaluated. The MLP model achieved an accuracy of 88.6% with balanced precision and recall. MLP-RF exhibited exceptional performance with high accuracy (99.0%), precision, recall, and F1 score. MLP-DT demonstrated strong performance with an accuracy of 98.9% and balanced precision and recall. MLP-kNN achieved a relatively high accuracy of 95.0% with good precision and recall. MLP-SVM showed moderate performance with an accuracy of 83.1%, relatively low recall, and moderate precision. Overall, the results indicate the effectiveness of MLP-RF, MLP-DT, and MLP-kNN in accurately detecting wheel defects, while MLP-SVM showed relatively lower performance.

These findings suggest the effectiveness of combining deep learning-based feature extraction with various machine learning models for accurate defect detection and classification. The achieved results highlight the potential of hybrid approaches in defect detection applications, showcasing the benefits of leveraging deep learning and traditional machine learning techniques in tandem. Future research can explore further enhancements to improve accuracy, explore different feature extraction methods, or investigate the applicability of the proposed system to other defect detection scenarios. One limitation of this study is the reliance on simulated data to train and evaluate the proposed model. While efforts have been made to create a realistic simulation model and generate comprehensive datasets, there may still be discrepancies between simulated data and real-world scenarios. It is important to validate the model’s performance using real-world data to ensure its effectiveness in practical applications.

To advance the field of defect detection and classification using hybrid deep learning methodologies, several avenues for future research can be explored. These include fine-tuning the pre-trained MLP model specifically for defect detection, investigating alternative deep learning architectures, such as CNNs or RNNs, and evaluating the effectiveness of data augmentation techniques. The integration of transfer learning and real-time implementation in real-time settings should also be considered. Additionally, validation on diverse datasets, deployment in real-time environments, and integration with automated decision-making systems are promising areas for further investigation. To achieve practical implementation, the proposed model can be effectively deployed on an edge computing platform. This deployment enables the real-time detection of wheel defects, providing prompt and accurate identification of any issues. By leveraging the computational capabilities of the edge computing platform, the model can efficiently process the incoming data from sensors and rapidly analyze them for the presence of wheel defects. This approach ensures that potential defects are promptly detected, and appropriate actions can be taken to maintain operational safety and efficiency in railway transportation.

## Figures and Tables

**Figure 1 sensors-23-06248-f001:**
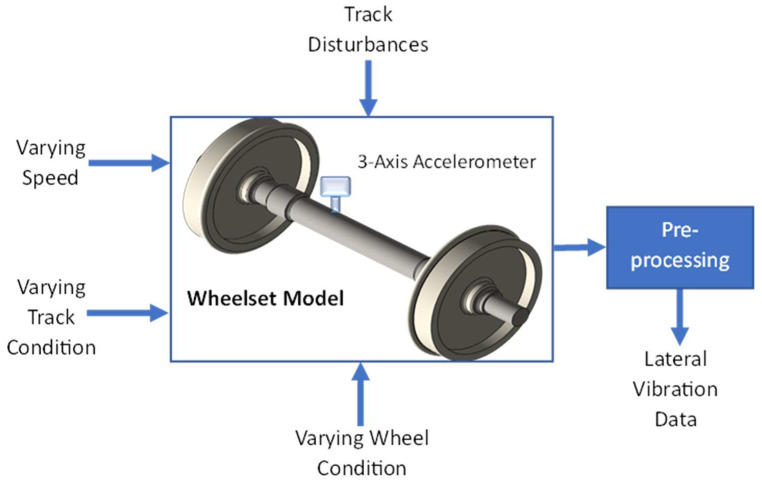
Data collection process.

**Figure 2 sensors-23-06248-f002:**
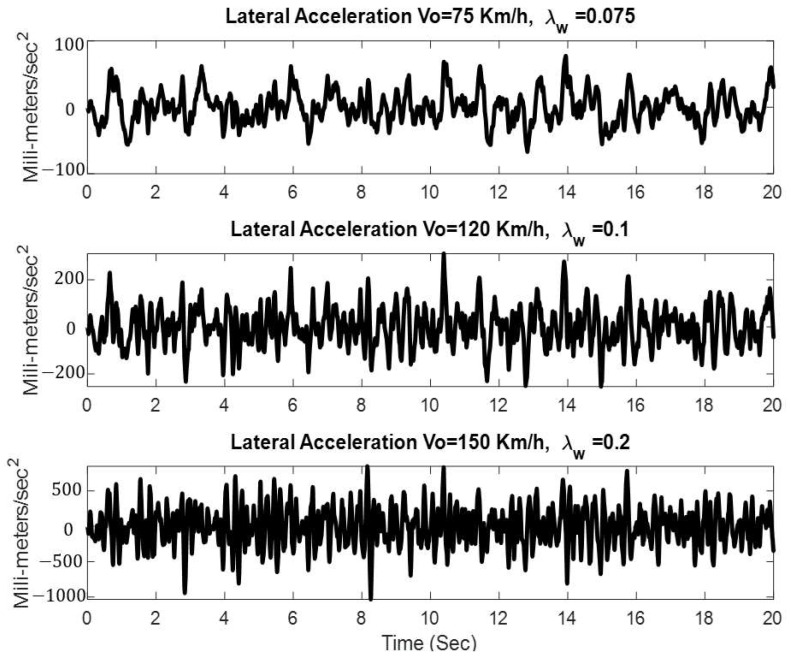
Lateral acceleration of the wheelset in different conditions.

**Figure 3 sensors-23-06248-f003:**
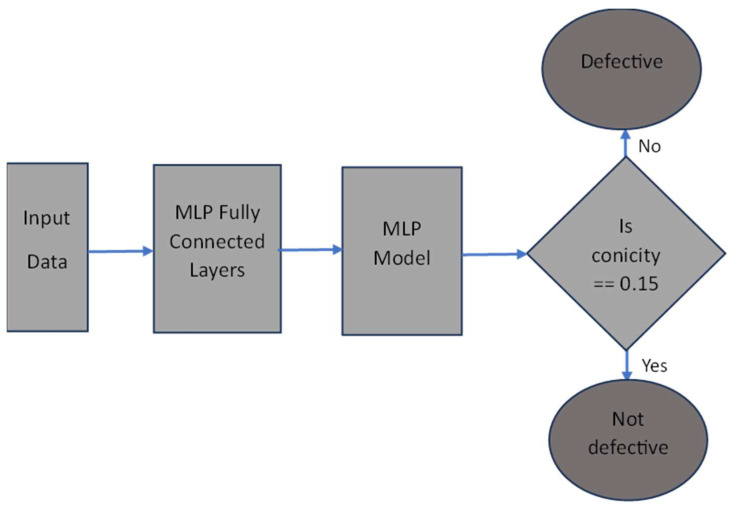
Model architecture.

**Figure 4 sensors-23-06248-f004:**
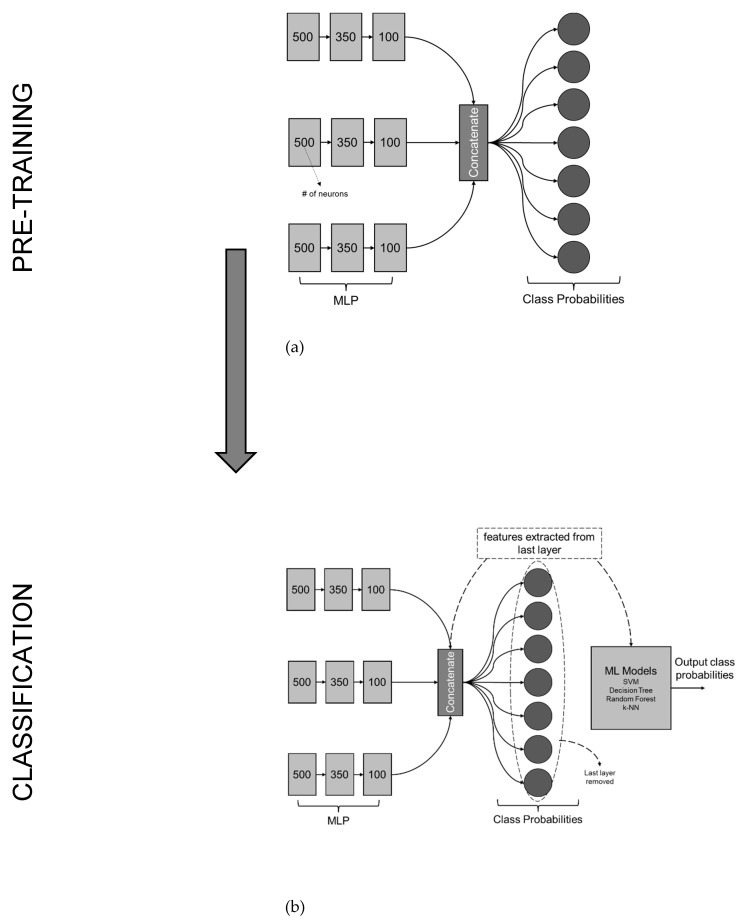
(**a**) Pre-training process. (**b**) MLP Network pruned with ML-based classifier. Once the MLP has been pre-trained and equipped with the ability to extract relevant features, these extracted features serve as inputs to the subsequent machine learning model, as shown in (**b**). The machine learning model leverages the extracted features to perform the final classification of conicity values. By utilizing the comprehensive and representative features obtained from the MLP, the machine learning model can make accurate predictions regarding the severity of defects in railway wheelsets.

**Figure 5 sensors-23-06248-f005:**
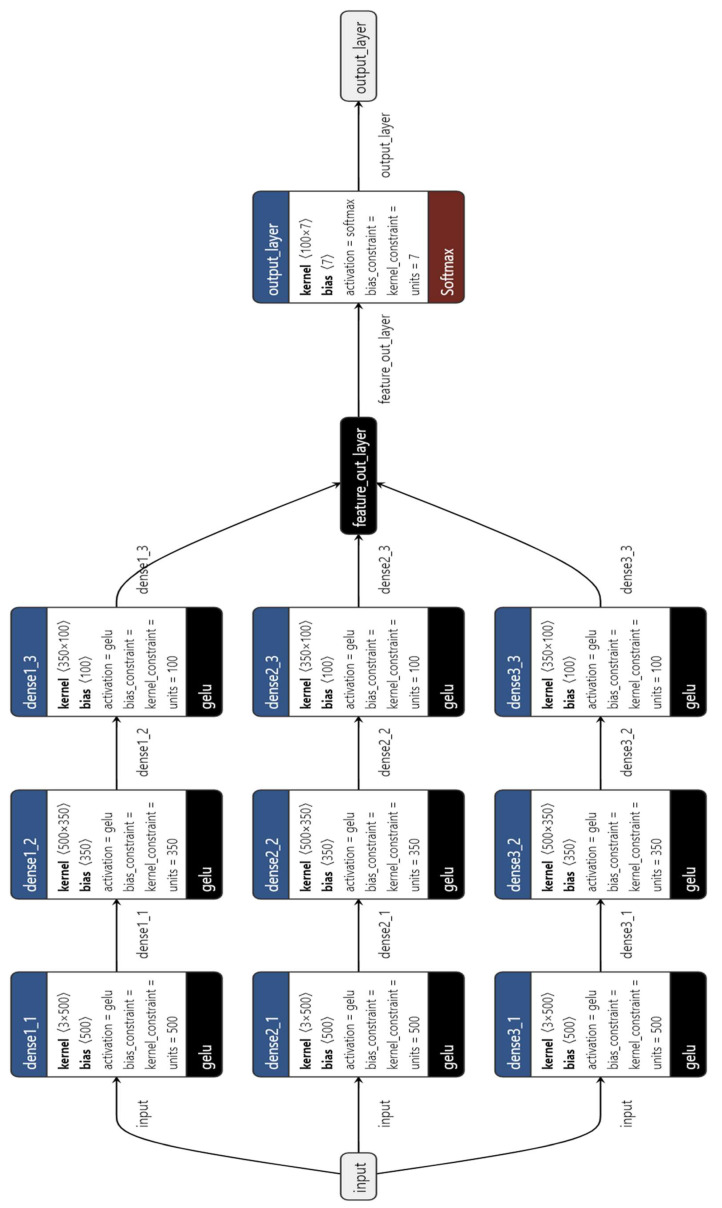
MLP networks’ architecture with three parallel branches and concatenation layer.

**Figure 6 sensors-23-06248-f006:**
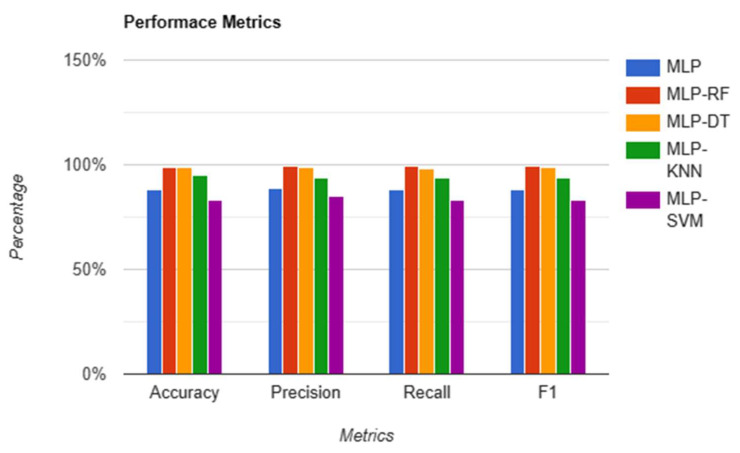
Bar chart of performance metrics of MLP, SVM, RF, DT, and k-N.

**Figure 7 sensors-23-06248-f007:**
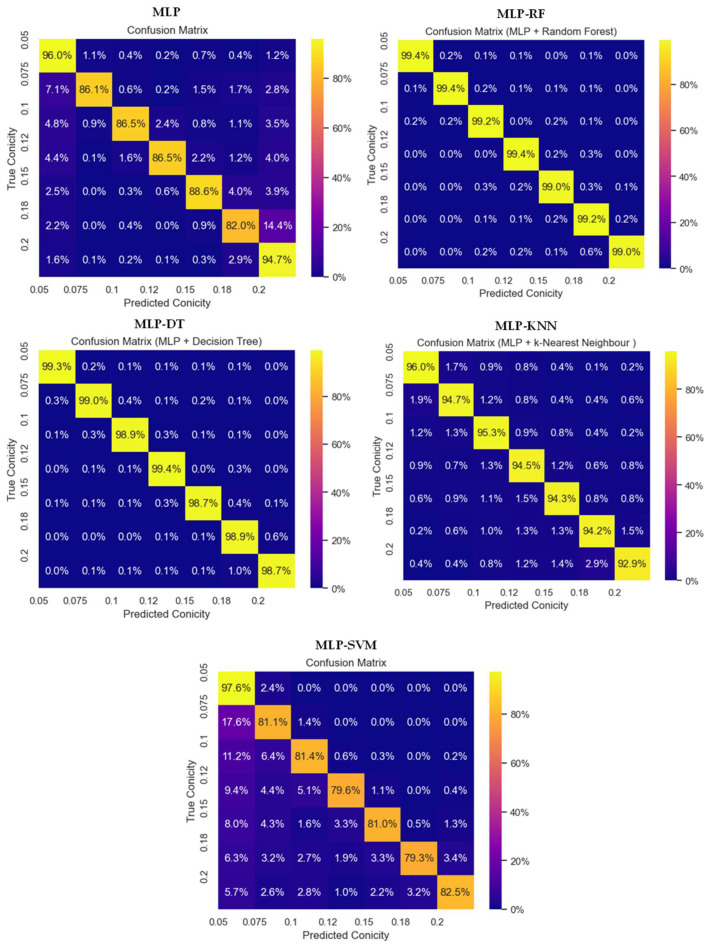
Classification results.

**Table 1 sensors-23-06248-t001:** MLP training summary.

Total Parameters	4471
Trainable parameters	4471
Non-trainable parameters	0

**Table 2 sensors-23-06248-t002:** Hyperparameters.

Hyperparameters	
Loss function	Categorical cross entropy
Activation function	GELU
Optimization algorithm	Adam
# of training epochs	100
Batch size	100

**Table 3 sensors-23-06248-t003:** Training procedure.

Step	Step Name	Details
1	Data preparation	Prepare and arrange the entire dataset with the number of columns as the number of features and the output label as the last column. The number of rows is the number of datapoints.
2	Pre-training phase	An MLP network is trained as a classifier first with inputs as feature vectors and output as conicity labels. The model is trained for 50 epochs using the GELU activation function and ADAM optimizer.
3	Feature extraction	The trained model in Step 2 is then used where the last classification layer is removed, giving us the concatenated layer. The output of this layer is of size 3 × 100. This is a feature matrix from the pre-trained MLP network.
4	ML model training	Multiple ML models, such as SVM, DT, RF, and kNN, are then trained on the extracted feature matrix from Step 3.

**Table 4 sensors-23-06248-t004:** Performance comparison.

Model	Performance Metrics			
	Accuracy	Precision	Recall	F1 Score
MLP	88.6%	89.6%	88.6%	88.7%
MLP-RF	99.0%	99.2%	99.2%	99.1%
MLP-DT	98.9%	98.9%	98.0%	98.9%
MLP-KNN	95.0%	94.5%	94.5%	94.0%
MLP-SVM	83.1%	85.6%	83.1%	83.5%

RF = random forest, DT = decision tree; KNN = k-nearest neighbor, SVM = support vector machine.

**Table 5 sensors-23-06248-t005:** Summary of results.

Model	Results Interpretation
MLP	This model achieved an accuracy of 88.6%, which means it correctly classified 88.6% of the defects. The precision of 89.6% indicates that when it predicted a defect, it was correct 89.6% of the time. The recall of 88.6% indicates that it identified 88.6% of the actual defects. The F1 score, which considers both precision and recall, is 88.7%.
MLP-RF	This model achieved high accuracy of 99.0%, indicating that it performed exceptionally well in classifying defects. The precision and recall of 99.2% suggest it had a very low rate of false positives and false negatives. The F1 score of 99.1% reflects the overall effectiveness of the model in detecting defects.
MLP-DT	This model achieved an accuracy of 98.9%, indicating strong performance in defect detection. The precision of 98.9% suggests that it had a very low rate of false positives. However, the recall of 98.0% indicates it missed a small portion of actual defects. The F1 score of 98.9% reflects a good balance between precision and recall.
MLP-kNN	This model achieved an accuracy of 95.0%, indicating a relatively high performance in detecting defects. The precision and recall of 94.5% suggest a low rate of false positives and false negatives. The F1 score of 94.0% reflects a good overall performance, though slightly lower than the previous models.
MLP-SVM	This model achieved an accuracy of 83.1%, which indicates it had a moderate level of performance in detecting defects. The precision of 85.6% suggests it had a relatively low rate of false positives. However, the recall of 83.1% indicates it missed a significant portion of actual defects. The F1 score of 83.5% reflects the overall effectiveness of the model, considering both precision and recall.

## Data Availability

All data, models, and codes generated or used during the study are available on request.

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
