# Peer review of "Wheel Defect Detection Using a Hybrid Deep Learning Approach"

_sensors, 2023, doi:10.3390/s23146248_

Round 1
Reviewer 1 Report

Moderate editing of English language required.
Reviewer 2 Report
This research paper presents a deep learning-based approach for the detection of various types of wheel defects using accelerometer data. A realistic simulation model of the railway wheelset is developed to generate a comprehensive dataset simulating faulty and non-faulty conditions. The generated data is then utilized to train and evaluate a hybrid deep learning model, employing a Multi-Layer Perceptron (MLP) as a feature extractor and multiple machine learning models (Support Vector Machine, Random Forest, Decision Tree, and k-Nearest Neighbors) for performance comparison. The proposed approach aims to enhance the accuracy of wheel defect detection while considering the cost-effectiveness and efficiency of onboard techniques. The contributions of this research work include the development of a realistic simulation model, the evaluation of sensor layout effectiveness, and the application of deep learning techniques for improved wheel flat detection.
I have some major revisions in the following
1- Proofreading is required to improve the presentation and correct types.
2- The main objectives and motivation of the study should be clarified.
3- More discussion is required for methodology and results also comparison.
This research paper presents a deep learning-based approach for the detection of various types of wheel defects using accelerometer data. A realistic simulation model of the railway wheelset is developed to generate a comprehensive dataset simulating faulty and non-faulty conditions. The generated data is then utilized to train and evaluate a hybrid deep learning model, employing a Multi-Layer Perceptron (MLP) as a feature extractor and multiple machine learning models (Support Vector Machine, Random Forest, Decision Tree, and k-Nearest Neighbors) for performance comparison. The proposed approach aims to enhance the accuracy of wheel defect detection while considering the cost-effectiveness and efficiency of onboard techniques. The contributions of this research work include the development of a realistic simulation model, the evaluation of sensor layout effectiveness, and the application of deep learning techniques for improved wheel flat detection.
I have some major revisions in the following
1- Proofreading is required to improve the presentation and correct types.
2- The main objectives and motivation of the study should be clarified.
3- More discussion is required for methodology and results also comparison.
Reviewer 3 Report
The authors explain and discuss their findings very clearly based on method description and figures.
this topic is very important and the paper is well-organized.
I think it could be accepted in this version.
Minor:
1. There are numerous grammatical errors throughout the manuscript.
2. The key words are not accurate and representative.
3. The format of the references should be standardized.
The authors explain and discuss their findings very clearly based on method description and figures.
this topic is very important and the paper is well-organized.
I think it could be accepted in this version.
Minor:
1. There are numerous grammatical errors throughout the manuscript.
2. The key words are not accurate and representative.
3. The format of the references should be standardized.
Reviewer 4 Report
Major changes:
Figure 2, the condition should be clear. “Is conicity == 0.05” -> Yes/No. Is it not important that is above or below this value?
Figure 4 is hard to read in this form. Please change it.
The formatting of tables is different, please use one format – regards to the journal template.
Minor changes:
Chapter 2.2. should not be in the last line of the page, I suggest putting on the next page.
Figure 2, the title should present model architecture.
Table 1, please use a dot rather than a comma.
Figure 5 – each value is higher than 60%, so starting the y axis from 60% will be better. The difference between bars will be easier to compare.
Figure 6, please delete the border of a table, and the title should be Classification Matrix.
Chapter 6 should not be in the last line of the page, I suggest putting into next page.
The English translation is fine, but there are small mistakes.
Round 2
Reviewer 1 Report
The paper can be accepted at the current form, but recommended the paper to go through an English editing service.
It is recommended that the paper uses an English editing service.